# Flavonoid Accumulation in an Aseptic Culture of Summer Savory (*Satureja hortensis* L.)

**DOI:** 10.3390/plants11040533

**Published:** 2022-02-16

**Authors:** Darya A. Khlebnikova, Evgeniya M. Efanova, Nina A. Danilova, Yaroslava V. Shcherbakova, Irina Rivera Sidorova

**Affiliations:** Department of Biotechnology, Russian State Agrarian University, Moscow Timiryazev Agricultural Academy, Timiryazevskaya 49, 127434 Moscow, Russia; evgeniya.sk.tt@mail.ru (E.M.E.); dan.nika2016@yandex.ru (N.A.D.); rosya3093@gmail.com (Y.V.S.); irinarivera@gmail.com (I.R.S.)

**Keywords:** *Satureja hortensis*, in vitro culture, callus, flavonoids, light-emitting diodes, spectral light composition

## Abstract

Summer savory (*Satureja hortensis* L.) is a medicinal and aromatic plant of the *Lamiaceae* family, a source of valuable secondary metabolites (monoterpenoids, rosmarinic acid, flavonoids). For this paper, flavonoid accumulation in an aseptic culture of summer savory was determined by using a colorimetric method. The organ specificity of flavonoid accumulation in aseptic plants was revealed: In leaves (8.35 ± 0.17 mg/g FW), flower buds (7.55 ± 0.29 mg/g FW), and calyx (5.27 ± 0.28 mg/g FW), flavonoids accumulated in significantly higher amounts than in stems (1.50 ± 0.22 mg/g FW) and corolla (0.78 ± 0.12 mg/g FW). We found that primary callus tissue formed from cotyledon and hypocotyl explants retains the ability to synthesize flavonoids at deficient levels (0.50 ± 0.09 mg/g FW and 0.44 ± 0.11 mg/g FW, respectively), that remained stable throughout six subcultures. Placing the callus tissue in monochrome lighting conditions with blue, green, and red light-emitting diode (LED) lamps leads to morphological changes in the tissue and decreased flavonoid accumulation compared to fluorescent lamps.

## 1. Introduction

Plants remain an indispensable raw material for the light and food industry, and a source of many valuable bioactive molecules/pharmacophores for humans [1,2]. The market for herbal medicines is rapidly growing; more than a quarter of all pharmaceuticals in industrialized countries are of herbal origin. According to a BBC report, the herbal medicines market will grow from $29.4 billion in 2017 to about $39.6 billion by 2022, with an annual growth rate of 6.1% [3]. However, technologies for industrial production of valuable medicinal molecules or substances based on plant cell and tissue cultures are not as widespread as the production of economically beneficial metabolites and cell biomass for the cosmetics and food industry; despite the fact that the need for these technologies increases from year to year [4]. The development of such biotechnological processes is based on extensive knowledge of plant physiology and biochemistry under in vitro conditions, molecular biology, and bioengineering. Thus, expanding the knowledge base for the characteristics of in vitro cultivation of medicinal, aromatic, and valuable agricultural plants, especially their cell cultures, contributes to the development of this industry sector and is an urgent task to research [4].

Summer savory (*Satureja hortensis* L.) is a herbaceous essential oil plant of the mint family (Lamiaceae). It is used in the traditional medicine of the Middle East, evidence-based medicine, cosmetology, and the food industry as condiments. Summer savory extracts and essential oils have a wide range of biological activities—antimicrobial, antiviral, antioxidant, fungicidal, antinociceptive, antitumor, hypoglycemic, hyperlipidemic, inhibition of amyloid beta protein aggregation, etc. [5]. The main components of the phytochemical profile are the monoterpenoids of the *p*-menthane group (thymol, carvacrol, *p*-cymene, and *γ*-terpinen); in addition, the aerial parts of plants are rich in flavonoids and contain rosmarinic acid [5,6].

Studies have revealed some biological activities associated with the accumulation of flavonoids and phenylpropanoids of summer savory. Mchedlishvili et al. (2005) found that the flavonoid fraction of *S. hortensis* lowers serum cholesterol levels in rabbits [7]. Accumulation of phenolic compounds (flavonoids and rosmarinic acid) determines the hypoglycemic activity of extracts; studies of local plant populations were conducted in Georgia; as a result, the drug Saturin was developed [6]. The capsule contains an aqueous extract and fine leaf powder. The drug is registered with the Ministry of Labour, Health and Social Welfare of Georgia as a drug for the treatment of diabetes mellitus type 2 [6,8]. Summer savory is actively used in traditional (for the Middle East) and evidence-based medicine. This was the reason for studying its qualitative and quantitative flavonoid content by research teams from different countries like Georgia, Serbia, Romania, Turkey, Finland, Greece, and Russia [6,7,8,9,10,11,12,13,14,15,16,17]. However, there are no data on the flavonoid accumulation in the aseptic culture of summer savory.

Light is an essential abiotic elicitor that affects many physiological processes in the plant, determining growth and development features [18]. Thus, changes in plant morphology and increased production of secondary metabolites in response to the use of different spectrum light sources have been studied in several species [19,20,21,22,23]. The response to different lighting regimes is often caused due to the fact that flavonoid biosynthesis occurs via the phenylpropanoid pathway and is regulated by numerous enzymes (phenylalanine ammonia lyase (*PAL*), chalcone synthase (*CHS*), chalcone isomerase (*CHI*), and flavonol synthase (*FLS*) genes, etc.), whose activity is both directly and indirectly induced by lighting parameters like spectral composition and intensity [24,25]. Cultivation under monochromatic light sources makes it possible to reveal the influence of one part or another of the spectrum on the course of flavonoid biosynthesis. Thus, cultivation under blue LEDs (460–470 nm) resulted in increased flavonoid accumulation in several species compared to other monochrome light modes (green, red, and yellow) in *Anoectochilus roxburghii* (Wall.) Lindl. [26], *Pisum sativum* L. [27], and *Glycine max* L. Merr. plantlets [28]. The combination of blue and red light also led to an increase in flavonoid accumulation in *Anoectochilus roxburghii* due to the activation of *CHI* and *FLS* genes expression [29].

Thus, tissue and cell culture of *S. hortensis*, which has retained the ability to synthesize valuable secondary metabolites characterized by intact plants, can be a raw material for the food industry in producing functional food additives and medicines. All of the above became the reason for our interest in studying summer savory aseptic cultures and the peculiarities of accumulation of biologically active secondary metabolites (flavonoids) in them.

## 2. Results and Discussion

### 2.1. Flavonoid Accumulation in Aseptic Plants

*In vitro cultivation of S. hortensis aseptic plants.* Aseptic plantlets of *S. hortensis* were cultivated into test tubes for 30 days; by the end of the initial culture, the plants reached a height of 6.5–7.5 cm and formed 3–5 pairs of true leaves (Figure 1a). Initial culture plants were divided into segments of 2–3 nodes and transplanted for further cultivation. By the end of the 1st subculture, the plants reached a height of 11–13 cm, formed 7–9 nodes and 3–5 lateral shoots. Plants of the 1st subculture developed from segments containing one or two nodes of the upper, middle, or lower parts of the initial culture plants, and after 30 days of cultivation reached a height of 15–20 cm, formed 6–9 nodes and 2–6 lateral shoots, the root system was actively developing. Most plants formed flower buds at the end of the 2nd subculture (Figure 1b) regardless of the segment origin at cutting (nodes of the upper, middle, or lower part of the mother plant). Flowering in plants of the 3rd subculture began after an average of 10 days of cultivation, and by the end of the subculture period, most plants were in the phase of mass flowering (Figure 1c,d). In contrast to Pistelli et al., 2013, we cultivated aseptic plants of summer savory on MS medium without auxins, since root formation was active and the addition of auxins was not necessary (Figure 1a,d) [30].

*Flavonoid content in S. hortensis aseptic plants.* The total flavonoid content in different organs was measured during three subcultures to study the characteristics of flavonoid accumulation in *S. hortensis* aseptic plants (Table 1).

Thus, in leaves of aseptic plants, the flavonoid content was 24 times higher than in stems at the end of the initial culture. By the end of the 2nd subculture, it increased insignificantly in leaves, while in stems, it increased almost fivefold; high flavonoids in flower buds were also noted. At the end of the 3rd subculture, there was a significant flavonoid decrease in leaves (by 16%) and an insignificant increase in stems (by 5%); in the corolla, the flavonoid content did not differ significantly from that in stems. In the calyx, it was noticeably inferior to the content in leaves (by 25%) and significantly superior to the content in stems and corolla (by 70% and 85%, respectively).

The total flavonoid content in flower buds of the 2nd subculture plants is higher in comparison to the content in stems. This can be explained by the accumulation of flavonoids in the photosynthetic part of the flower–calyx, and not in the forming corolla. Thus, the flavonoid accumulation in *S. hortensis* aseptic plants occurs mainly in leaves, flower buds, and calyx; much less is accumulated in stems and corolla of the flower. The flavonoid contents in leaves did not change during cultivation in three subcultures. The organ-specific flavonoid accumulation we observed in *S. hortensis* aseptic plants has been noted by many researchers for several species [31,32].

The study of the flavonoid accumulation in aseptic culture of summer savory has not been previously carried out, however, there is data on the flavonoid content in plants grown under in vivo conditions. According to Masković et al., 2017, the flavonoid content in *S. hortensis*, depending on the extraction method, can vary from 5.23 ± 0.76 to 28.42 ± 0.29 mg/g of dry extract in routine equivalent [13]. Summer savory growing in Georgia accumulated 1.6–1.7% of flavonoids and phenolcarboxylic acids in air-dry leaves and flowers collected during budding and flowering [6].

### 2.2. Flavonoid Accumulation in Callus Tissue

*Induction and cultivation of S. hortensis callus tissue.* Two types of explants—cotyledon leaves and hypocotyls—were cultured on an MS medium with the addition of 1 mg/L BAP to obtain primary callus. The callus formed a light green color with small yellow areas, heterogeneous in density; looser and denser areas could be distinguished; shoot and root organogenesis took place actively (Figure 2).

After 2–3 weeks of explant cultivation, meristematic foci were formed on the callus tissue sites, and small shoots of normal morphology, hyperhydric teratomas, and roots were formed due to organogenesis.

*Flavonoid content in callus tissue of S. hortensis.* The total flavonoid content in the primary callus after 30 days of cultivation (initial culture) was significantly lower than in the leaves and flower buds of aseptic plants: in callus from cotyledon 0.50 ± 0.09 mg/g FW; in callus on hypocotyl explants 0.44 ± 0.11 mg/g FW. This may be due to the need for cell differentiation for flavonoid biosynthesis [31,32,33]. In favor of this hypothesis, the flavonoid content in regenerants (the shoots of normal morphology and hyperhydric teratomas were analyzed together) formed on the primary callus was 1.26 ± 0.21 mg/g FW.

The type of explant (cotyledon or hypocotyl) did not affect the flavonoid accumulation in the primary callus formed on it. Therefore, cotyledon leaves and hypocotyls can be used as explants to obtain callus culture for flavonoid synthesis.

The callus formed on hypocotyl explants was chosen for further experimentation because visually, the growth of this kind of callus tissue was more active than in cotyledon explants. Primary callus tissue areas were cultured for six subcultures. The callus appearance and consistency changed insignificantly; the tissue remained green with light yellow areas; the callus consistency was quite dense, easily to split into smaller aggregates; no organogenesis was observed (Figure 3).

The data on changes in contents of total flavonoids in the callus tissue over six subcultures are shown in Figure 4.

Having analyzed the data of Figure 4, we can draw the following conclusions. The flavonoid accumulation in callus tissue is much lower than in aseptic plants. This trend is typical for the primary callus tissues of some species, especially in the absence of elicitation [31]. Throughout six subcultures, there were no significant deviations in flavonoid accumulation from the primary callus (content did not exceed 0.6 mg/g FW). Small fluctuations in flavonoid accumulation during the subculture period can be explained by the genetic and morphologic heterogeneity of callus tissue.

### 2.3. Effect of Monochromatic Light on Callus Tissue Characteristics and Flavonoid Content

To study the effect of the light regime on flavonoid accumulation, were chosen the following conditions: LED lamps of a monochrome spectrum–blue, green and red.

*Appearance and consistency of callus tissue.* Callus of the 4th subculture obtained from hypocotyl explants was used for the experiment. Cultivation under blue LEDs during the 5th subculture resulted in the growth of callus tissue of green-yellow color with slight red spots; the callus consistency was slightly watery and friable; it disintegrated into small aggregates, with quite dense areas (Figure 5a). Cultivation under green LEDs led to the formation of an actively growing soft, easily disintegrating into small aggregates, watery callus from a light-yellow color to green. In isolated cases, rhizogenesis was observed; thin roots 2–3 mm in length were formed (Figure 5b). Red light promoted growth of green-yellow callus with small brown areas, with denser consistency than under green light (Figure 5c).

*Flavonoid accumulation in S. hortensis callus tissue under the monochromatic light influence.* Cultivation during the 5th subculture under monochromatic LED lamps did not result in significant changes in flavonoid accumulation (Figure 6).

Summarizing the data obtained, we can conclude that the appearance of callus tissue when cultured under monochrome LED light: blue, green, and red, does not differ significantly, the most actively growing callus under green light is characterized by low levels of flavonoid accumulation, the same as the callus under the other monochrome lamps. Thereby, the factor of lighting conditions in the selected modes did not affect the accumulation of flavonoids in S. hortensis callus tissue samples. However, according to some researchers, the cultivation of aseptic plants under monochrome lighting affected flavonoid accumulation [31]. It may be due to disruption of both the genetic material in the cell and disorders of its expression, which are characteristic of callus tissue as a heterogeneous structure [31].

A number of researchers note an increase of flavonoid accumulation during the cultivation of aseptic plants and cell cultures under blue light [34]. For example, this trend is typical for Saccharina japonica (J.E. Areschoug) C.E. Lane, C. Mayes, Druehl & G.W. Saunders, Saussurea medusa Maxim., and Lactuca sativa L. [34,35,36,37]. In 2018, Li et al. studied the influence of the monochrome spectrum of LEDs, as well as the photoperiod and radiation intensity on the growth of the embryogenic callus of Dimocarpus longan Lour. The following dependence of the flavonoid accumulation on the spectrum of LED lamps was noted–blue light > green light > dark > white light > red light [34].

## 3. Materials and Methods

**Plant material.** *S. hortensis* ‘Gnom‘ seeds were used for introduction to in vitro culture. ‘Gnom’ (seed produced by the Agrogroup Biotechnica)—a compact form of summer savory, forming a strongly branching bush 10–12 cm high.

**Introduction of *S. hortensis* to in vitro culture and further cultivation of aseptic plants.** To obtain aseptic plantlets, *S. hortensis* seeds were treated with 5% sodium hypochlorite solution (NaOCl) for 10 min, then washed in two portions of sterile distilled water and placed in Petri dishes with hormone-free Murashige and Skoog (MS) medium with 3% (*w*/*v*) sucrose, 0.8% (*w*/*v*) agar [38] for germination. After 8–10 days, the seedlings were transplanted into test tubes with a medium of the same mineral composition. After 30 days of cultivation, the plants were divided into segments of 2–3 nodes and transplanted into 0.9-L glass culture bottles covered with cotton-gauze plugs, 3–4 pieces per bottle with the nutrient medium of the same composition for more intensive biomass accumulation. Aseptic plants were cultured in a growth room with a 16-h photoperiod, a temperature of 21 ± 2 °C, under white fluorescent lamps with an illuminance of 2500 Lx, photon flux density 22 ± 2.2 μmol/(s∙m^2^), 4000K (manufacturer ‘OSRAM‘, Smolensk, Russia). The duration of the subculture was 30 days.

**Induction of callus formation.** Cotyledons and hypocotyls of seedlings at the age of 6–8 days placed in Petri dishes with an MS medium supplemented with 1 mg/L benzylaminopurine (BAP) were used to induce callus formation. Primary callus tissues 0.3–0.5 cm in diameter were transplanted after 30–35 days to the medium of the same mineral and hormonal composition and in vitro culture was continued. The duration of subsequent subcultures was 25–30 days. Callus tissue in vitro culture was performed under the growth room conditions indicated above. Three samples of callus tissue were taken from five Petri dishes at the end of each subculture to analyze the total flavonoid content.

**Callus culture under monochrome LED lighting.** Petri dishes with callus tissue were placed under light regimes with the following characteristics: blue light (λ_max_ = 460 nm) (Figure 7a), green light (λ_max_ = 520 nm) (Figure 7b), and red light (λ_max_ = 660 nm) (Figure 7c), to study the monochrome light effect. The photon flux density in all variants was 70 ± 10% μmol/(s∙m^2^).

**Preparation of extracts.** A sample of fresh plant material (callus samples or aseptic plant segments collected at the end of the subculture) was ground in 96% ethyl alcohol solution, left in the refrigerator for 48 h for extraction at 4 ± 2 °C, filtered, and used for analysis.

**Determination of the total flavonoid content.** We used a colorimetric analysis to determine the total flavonoid content (Varian Cary^®^ 50 UV-Visible spectrophotometer, Varian Inc., Palo Alto, California, USA). Preparation of the mixture for spectrophotometry: 1000 µL ethanol extract, 50 µL 10% aluminum chloride (AlCl_3_) in ethanol solution, 50 µL potassium acetate (CH_3_COOK), and 1400 µL distilled water was kept for 30 min. Optical density was measured at a wavelength of 415 nm. The calibration curve was made by quercetin.

**Statistical processing of the data obtained.** Statistical processing of the experimental results was performed using the statistical functions of Excel (Microsoft Office) and the AgCStat software package. The data in the tables and graphs are presented as the arithmetic mean of the samples ± confidence interval (*p* = 0.05). All experiments were performed in five biological and three analytical replicates.

## 4. Conclusions

Aseptic *S. hortensis* plants accumulate valuable secondary metabolites: flavonoids. The accumulation was found to occur mainly in leaves (8.35 ± 0.17 mg/g FW), flower buds (7.55 ± 0.29 mg/g FW), and calyx (5.27 ± 0.28 mg/g FW). Fluctuations of flavonoid accumulation in different organs of *S. hortensis* aseptic plants during three subcultures were also noted. The level of flavonoid accumulation in the resulting morphogenic callus tissue (0.44 ± 0.11 mg/g FW) and regenerants (1.26 ± 0.21 mg/g FW) was found to be significantly lower than in aseptic plants. The observed slight fluctuations in the level of flavonoid accumulation in the callus throughout six subcultures may be related to the genetic and morphological heterogeneity of the callus tissue. Callus culturing under monochromatic light affected the morphological features, but did not significantly change flavonoid accumulation. The total flavonoid content in callus simples under blue LEDs was 0.24 ± 0.07 mg/g FW, under green LEDs-0.18 ± 0.03 mg/g FW, and under red LEDs-0.19 ± 0.04 mg/g FW.

## Figures and Tables

**Figure 1 plants-11-00533-f001:**
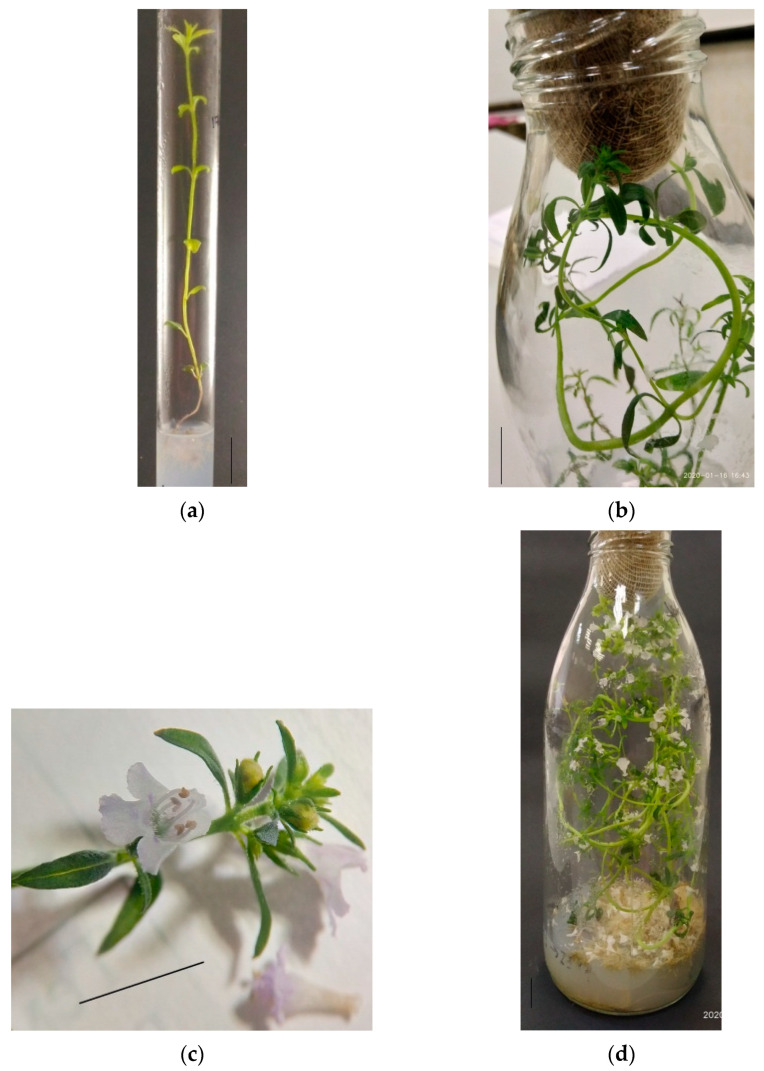
Appearance of *S. hortensis* aseptic plants: (**a**) initial culture, (**b**) 2nd subculture, (**c**) 3rd subculture, and (**d**) aseptic plant shoot of the 3rd subculture with flower buds and flowers. Bars—0.5 cm.

**Figure 2 plants-11-00533-f002:**
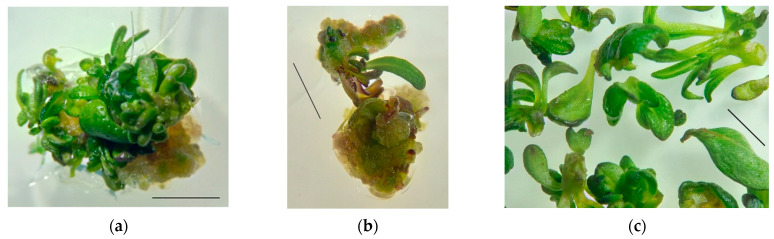
Appearance of the primary callus and regenerant plants of *S. hortensis*: (**a**) callus from cotyledon leaves, (**b**) callus from hypocotyls, and (**c**) regenerant plants. Bars—0.5 cm.

**Figure 3 plants-11-00533-f003:**
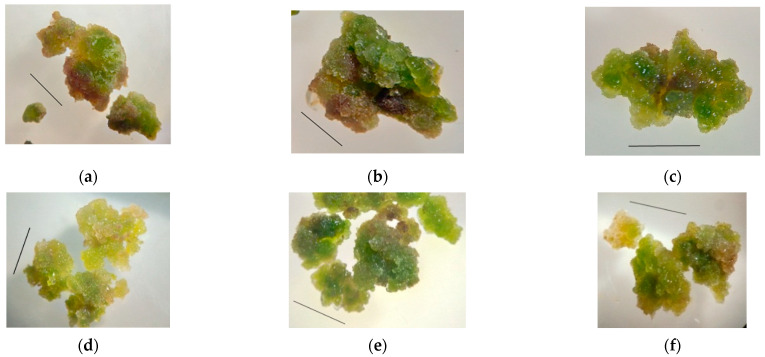
Changes in the appearance of *S. hortensis* callus tissue over six subcultures: (**a**) 1st subculture, (**b**) 2nd subculture, (**c**) 3rd subculture, (**d**) 4th subculture, (**e**) 5th subculture, and (**f**) 6th subculture. Bars—0.5 cm.

**Figure 4 plants-11-00533-f004:**
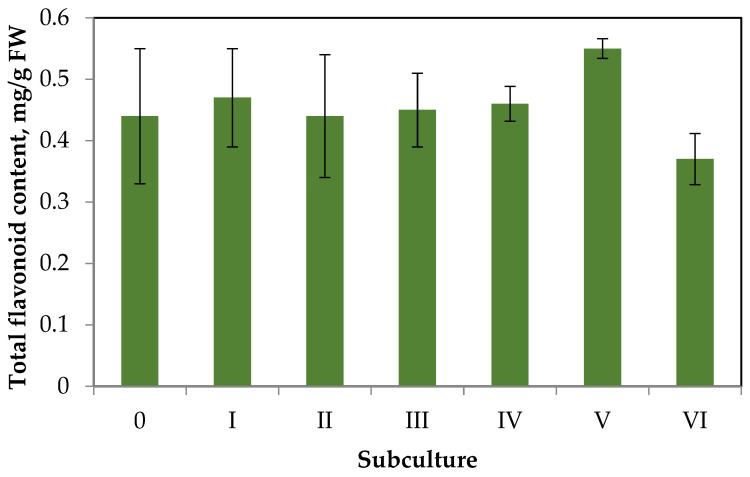
Dynamics of flavonoid accumulation in *S. hortensis* callus over six subcultures.

**Figure 5 plants-11-00533-f005:**
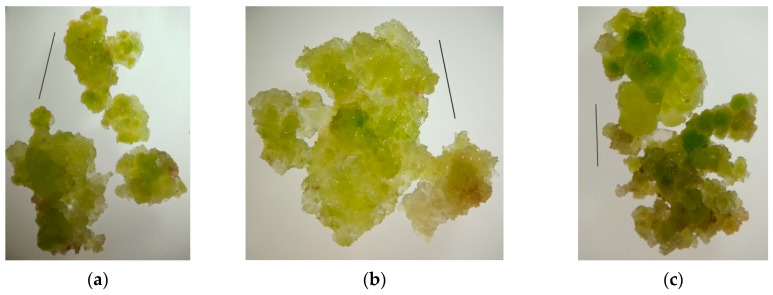
Appearance of callus tissue after cultivation under LED lamps of monochromatic spectrum: (**a**) blue LEDs; (**b**) green LEDs; and (**c**) red LEDs. Bars–0.5 cm.

**Figure 6 plants-11-00533-f006:**
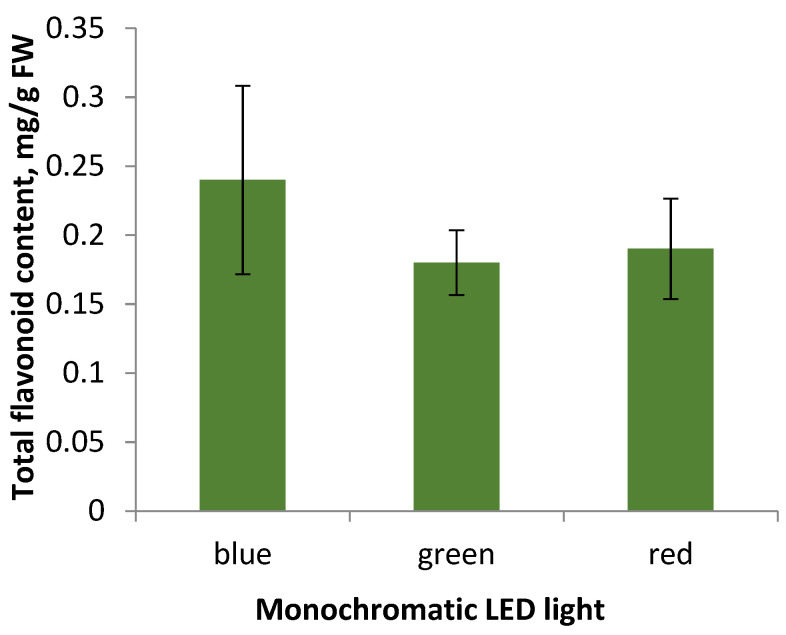
Total flavonoid content in callus tissue of the 5th subculture under different light conditions of cultivation.

**Figure 7 plants-11-00533-f007:**
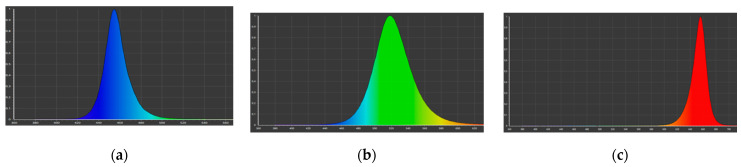
Monochrome LED lamps spectrum: (**a**) blue LEDs; (**b**) green LEDs; and (**c**) red LEDs.

**Table 1 plants-11-00533-t001:** Total flavonoid content in various parts of *S. hortensis* aseptic plants.

Plant Material	Subculture	Total Flavonoid Content, mg/g FW
Leaf	0	8.17 ± 1.07
Stem	0	0.34 ± 0.04
Leaf	1	8.20 ± 0.78
Stem	1	0.93 ± 0.15
Leaf	2	8.35 ± 0.17
Stem	2	1.50 ± 0.22
Flower bud	2	7.55 ± 0.29
Leaf	3	7.02 ± 0.90
Stem	3	1.58 ± 0.76
Corolla	3	0.78 ± 0.12
Calyx	3	5.27 ± 0.28

## Data Availability

Data is contained within the article.

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
