# Peer review of "Flavonoid Accumulation in an Aseptic Culture of Summer Savory (Satureja hortensis L.)"

_plants, 2022, doi:10.3390/plants11040533_

Round 1

Reviewer 1 Report

I think this is a paper that shows very interesting data. However, it is undeniable that the expression is redundant as a whole. I think that it is better to express the purpose more clearly.

  1.  Introduction  Although the authors touched on food production, I think that phytocemical substance production is the main purpose range in tissue culture systems. The current description can be misleading. In other words, I think that the research purpose has been expanded too much. I think it has enough content as an experiment for phytochemical production. The notation of the gene symbol is incorrect. It is an international rule that genes that function normally are written in italics in uppercase, and genes that lack function are written in italics in lowercase. Such errors can make your dissertation unreliable.
  2.  Results and discussion    It is common to set the control plot to white, but it is customary to arrange the monochromatic LEDs in order from the one with the shortest peak wavelength.
  3.  Materials and methods   First of all, the explanation of material plants is insufficient. You should describe what kind of plant it is, its own habitat and the conditions for basic cultivation because it is easy and good understanding for reader. Also, the cotton plug is used in the culture method, but the ventilation frequency of the culture vessel is a factor that greatly affects the culture result, so isn't the actual ventilation frequency measured? It is a point that should be measured and described. Also, fluorescent lamps are used when raising seedlings, but the manufacturer's model affects the subsequent reaction under the LED. This is also the point that should be desicribed. And, although the emission characteristics of the LED used are shown only at the peak wavelength, each LED lamps has a different emission spectrum, so it should be shown. Furthermore, since white LEDs have completely different emission characteristics depending on the phosphor used, the effect of light quality cannot be considered unless shown.      I request improvement in the above. 

Author Response

We would like to thank you for your careful reading our work. We have tried to amend the manuscript in accordance with your comments.
1. Introduction
We have shortened the introduction and corrected the spelling of genes.
2. Results and discussion. We have corrected the figure 6.
3. Materials and methods. We have described morphological features of the savory cultivar used in experiments, indicated the manufacturer of fluorescent lamps, and presented the monochrome LED lamp spectrum. As for the ventilation of culture vessels with cotton-gauze plugs - ventilation was not carried out.
We plan to correct errors in English by taking the opportunity to edit the manuscript by the publisher

Reviewer 2 Report

Dear authors,

The subject of the manuscript is interesting, but the text needs significant improvement. I suggest а major revision of the manuscript. The improvement of the English language is needed, concerning the grammar, stylistics and technical mistakes, as well.

In Abstract

Page 1 Line 10

In the recent years author abbreviation Martinov is not specified after Family name.

In Introduction

The introduction needs to be rewritten to have clear purpose and well-structured background information for the proposed study. The authors should focus on previous research on in vitro culture of Satureja hortensis L and accumulation of flavonoids in them. Although there are similar studies in the available literature, they have not been reported in the manuscript. There is a lot of superfluous information in introduction; for example description of individual compounds of monoterpenoids, flavones, flavanols, flavanones, flavonoid glycosides, flavonol glycosides, flavanone glycosides in S. hortenisis. In addition the authors should emphasize the need and novelty of their research on science.

In Results and Discussion

Page 3 line 118 The sentence „Aseptic plantlets of S. hortensis at the age of 8-10 days were transplanted into test tubes on a hormone-free MS medium and cultivated for 30 days“ is more suitable for Materials and Methods section.

I recommend using the term subculture instead passage. The term passage is more suitable for plant cell culture.

Page 5 line 154 The sentence „As a result, the higher the leaf on the aseptic plant, the more flavonoids are accumulated in them.“ is not clear

Page 6 line 203 I recommend the authors to discuss effect of monochromatic light (white, green and red LEDs) on the growth characteristics and obtained biomass of callus culture, not only the callus tissue appearance.

The Discussion is very poor and almost missing. Presented results are discussed with only 2 references.

In Materials and Methods

Page 9 Line 269

Authors should include the statistical differences at a given p-value.

Author Response

We would like to thank you for your careful reading our work. We have tried to amend the manuscript in accordance with your comments.
In the annotation, we removed "Martinov" after plant family writing.
1. Introduction
We have shortened the introduction and added information about in vitro culture of summer savory
2. Results and discussion. We have removed the overly detailed description of the cultivation process, we were using the term "subculture" instead "passage". Unfortunately, we cannot provide data on the growth characteristics and obtained biomass of callus culture, as no such measurements have been made. We will try to take this remark into account by carrying out further experiments. We have expanded the discussion and increased the bibliography.
3. Materials and methods. We have clarified the p-value.

We plan to correct errors in English by taking the opportunity to edit the manuscript by the publisher

Reviewer 3 Report

Khlebnikova and colleagues presented interesting results of a systematic study of the influence of cultivation conditions on the content of flavonoids. The quality of the studies performed is beyond doubt. However, it is worth adding the results of the qualitative determination of flavonoids by HPLC or HPLC MS, these results are important in terms of composition (apigenin, quercetin, mangefirin) and also the analysis of the presence of glycosylated forms.

Author Response

We would like to thank you for your careful reading our work and your positive feedback.
Unfortunately, we did not carry out a qualitative analysis of flavonoids by HPLC or HPLC MS. We have analysed the total flavonoid content using the colorimetric method.

We plan to correct errors in English by taking the opportunity to edit the manuscript by the publisher

Round 2

Reviewer 1 Report

If there is no description of the white LED information, there is insufficient information, and if this is not shown, the data cannot be judged, so be sure to add it. Others are minor technical term changes, but they will be easier to understand, so please consider modifying them.

Line 17, 102, 103,108, 125, 155, 160, 162-163, 165, 169, 171, 220, 231, 267, 270:subcultivation → subculture, subcultivations → subcultures

Line 29-32: These two sentences are descriptions of food (staple food) production that are not directly related to this research, and are unnecessary and deleted.

Line 42:food industry and cosmetology → cosmetics and food industry

Line 50: Family name is written in roman type.

Line 51: spices → condiment

Line 83: Glycine max (italic) L. → Glycine max (italic) (L.) Merr.

Line 95, 98, 105, 113, 144, : zero subculture → initial culture

Line 122: sepal  → calyx

Line 125: accumulation → contens

Line 129-130: What does layer mean? Is it the difference in the position of the leaves? It should be described more specifically and show actual data.

Line 165: Dynamics of flavonoid accumulation → Changes in contents of total flavonoids

Line 176: Why is the emission spectrum of white LEDs not shown? It should be shown as important information.

Line 181: a ligh-  → a light-

Line 182: grren  → green

Line 194: White LEDs are not monochromatic light sources. Change the expression.

Line 209: The author names are not included in the three scientific names. Everything else is in it and should be included.

Line 210: studed → studied

Line 217: S. hortensis (italic) seeds of the cultivar Gnom → S. hortensis (italic) ‘Gnom’seeds

Line 218: Cultivar Gnom → ‘Gnom’

Line 225-228: Did you add sucrose to the medium?

Line 236, 237,263: cultivation → in vitro culture

Line 240: cultivation  → culture

Line 240-244: Why is the emission spectrum of white LEDs not shown?

Author Response

Dear reviewer!
We would like to thank you for your recommendations and corrections. We have excluded data on white LEDs, because for technical reasons we cannot provide their spectrum. Line 29-32, 129-130 were dropped. The rest of the fixes you recommended have been done.

Reviewer 2 Report

Dear Authors and Editor,

I read the revised manuscript, as well as the author response file. I’m satisfied with the corrections. The Introduction and Results and Discussion have been improved. However I recommend the authors to include in the discussion previous studies of aseptic culture of (Satureja hortensis L.). For example:

Karimi, N., Ghasmpour, H. R., & Yari, M. (2014). Effect of different growth regulators on callus induction and plant regeneration of Satureja species. Annual Research & Review in Biology, 2646-2654.

Pistelli, L., Noccioli, C., D'Angiolillo, F., & Pistelli, L. (2013). Composition of volatile in micropropagated and field grown aromatic plants from Tuscany Islands. Acta Biochimica Polonica, 60(1).

Güllüce, M., Sökmen, M., Daferera, D. I. M. I. T. R. A., Aǧar, G., Özkan, H., Kartal, N. U. K. E. T., ... & Åžahin, F. (2003). In vitro antibacterial, antifungal, and antioxidant activities of the essential oil and methanol extracts of herbal parts and callus cultures of Satureja hortensis L. Journal of Agricultural and food chemistry, 51(14), 3958-3965.

Page 3 line 107 The subtitle „Flavonoid accumulation in S. hortensis aseptic plants“  is the same as that of page 2 line 93  and should be change to Flavonoid content in S. hortensis aseptic plants

Page 5 line 143 Flavonoid accumulation in callus tissue of S. hortensis. is the same as that of page 4 line 132  and should be change to  Flavonoid content in callus tissue of S. hortensis.

Kind regards and good luck!

Recommendation: Accept Submission

Author Response

Dear reviewer!
We would like to thank you for your recommendations and corrections. Page 3 line 107 and page 5 line 143 have been corrected. We used one of the sources you suggested for the results and discussion - Pistelli, L., Noccioli, C., D'Angiollo, F., & Pistelli, L. (2013) Composition of volatile in micropropagated and field grown aromatic plants from Tuscany Islands. Acta Biochimica Polonica, 60(1).

Round 3

Reviewer 1 Report

Although the authors are conducting very interesting research, it is a pity that they lack basic mastery of photochemical reactions.